# Hidradenitis Suppurativa in Patients with HIV: A Scoping Review

**DOI:** 10.3390/biomedicines10112761

**Published:** 2022-10-31

**Authors:** Laura Macca, Vittoria Moscatt, Manuela Ceccarelli, Ylenia Ingrasciotta, Giuseppe Nunnari, Claudio Guarneri

**Affiliations:** 1Department of Clinical and Experimental Medicine, Section of Dermatology, University of Messina, Italy C/O A.O.U.P. “Gaetano Martino”, via Consolare Valeria 1, 98125 Messina, Italy; 2Unit of Infectious Diseases, Department of Clinical and Experimental Medicine, University of Catania, Italy C/O ARNAS “Garibaldi”, “Nesima” Hospital, Via Palermo 636, 95122 Catania, Italy; 3Department of Diagnostics and Public Health, University of Verona, 37134 Verona, Italy; 4Unit of Infectious Diseases, Department of Clinical and Experimental Medicine, University of Messina, Italy C/O A.O.U.P. “Gaetano Martino”, via Consolare Valeria 1, 98124 Messina, Italy; 5Department of Biomedical and Dental Sciences and Morphofunctional Imaging, Section of Dermatology, University of Messina, Italy C/O A.O.U.P. “Gaetano Martino”, via Consolare Valeria 1, 98125 Messina, Italy

**Keywords:** hidradenitis suppurativa, human immunodeficiency virus, HAART, biologics, autoimmunity

## Abstract

Hidradenitis suppurativa (HS) is a chronic, debilitating skin disease of the apocrine glands. Bibliographic search revealed few studies concerning the association between HS and human immunodeficiency virus (HIV). To assess this link, we performed a systematic review of the current knowledge through a careful analysis of the relevant and authoritative medical literature in the field. Results showed that people with HIV are particularly susceptible to developing HS with the characteristic involvement of atypical sites, such as face or thighs, due to HIV-related immunosuppression. Based on the pathogenesis of both conditions and according to our review, we suggest that HIV screening should be routinely performed in suspected cases while monitoring and integrated approach in management are mandatory in the management of HIV-positive patients with HS.

## 1. Introduction

Hidradenitis suppurativa (HS) is an inflammatory follicular skin disease characterized by chronic, painful, deep-seated, inflamed lesions at the apocrine gland-bearing areas of the body (e.g., axillary, inguinal and anogenital regions) [1,2,3]. With a reported prevalence of up to 4%, HS is more than twice as common in young women compared to young men [1]. Although the pathogenic mechanism has still not been fully understood, it is believed that follicular hyperkeratosis causes plugging and dilation, resulting in hair follicle rupture [4,5]. This event is followed by a significant inflammatory response, with the recruitment of self-perpetuating inflammatory mediators [4,5].

Patients with HS are at high risk of infections because of the skin-barrier alterations coming from erosions and suppurating lesions, treatment with immunosuppressants, also for topical use, and comorbidities such as diabetes [6]. According with these potential hazards, serious infections may occur, including those due to the human immunodeficiency virus (HIV), that significantly influences the clinical course and presentation of such chronic skin conditions [7]. HIV is an RNA virus grouped to the family of Retroviridae, associated with slow onset of immunosuppressive diseases and neurologic disorders [8]. HIV tropism for CD4-expressing T-cells and myeloid cells is the major determinant in the pathogenesis of the disease. Indeed, an uninhibited viral replication causes a progressive reduction of CD4 T-cells count and, consequently, an insufficient ability to activate and control immune response [9]. The viral transmission requires direct exposition to infected blood or secretions and cutaneous/mucosal damage. A wide variety of behavioral and biologic risk factors were associated with the risk of transmission, including the frequency and amount of different sexual partners, the condoms use (or non-use), the use of hormonal contraceptives, immunologic status, the presence or absence of advanced HIV infection (AIDS), circumcision, and sexually transmitted diseases [10,11,12,13,14,15,16]. The acute burst of viremia and wide dissemination of virus in primary HIV infection seems to be associated with HIV acute syndrome [17]. As the natural history of the infection continues, acute infection is followed by a period of viral latency, normally lasting 3–10 years, after that the collapse of the immune system begins. Acquired Immunodeficiency Syndrome (AIDS) is defined as the last stage of HIV infection (stage C) and usually develops 8–10 years after primary infection in the absence of treatment. AIDS may be manifested in several different ways, including lymphadenopathy and fever, opportunistic infections, malignancies, and AIDS-related dementia [18]. The causative agents of the secondary infections are usually opportunistic organism such as P. Jirovecii, atypical mycobacteria, T. Gondii, Candida, CMV and other organisms that do not ordinary cause disease in the absence of a compromised immune system. However, secondary infections can also be caused by common bacterial and mycobacterial pathogens [19,20,21]. With the advent of highly active combination antiretroviral therapy in the mid-1990s and routine use of antimicrobial prophylaxis, the rates of AIDS defining opportunistic illnesses among HIV-infected adults and children have declined dramatically in the industrialized countries [22,23]. Nonetheless, opportunistic illnesses remain a leading cause of hospitalization and death among HIV-infected persons. HIV patients are more easily subject to cancers, rheumatological and skin disorders than the seronegative population. Moreover, the clinical course of this diseases is often associated with high morbidity and mortality [22,23].

To assess the association between HIV and HS, we performed a systematic review of the current literature on the topic and discussed the results.

## 2. Materials and Methods

With the purpose of reviewing the pertinent literature, we checked the PubMed (https://ncbi.nlm.nih.gov/PubMed, accessed on 14 April 2022), Scopus and Web of Science databases using the search string “Hidradenitis suppurativa” [All Fields] AND “HIV” [All Fields] OR “AIDS” [All Fields], without time limits. Using the PRISMA flowchart for the drafting of our systematic literature, we read the abstracts of each article whose title suggested the association between hidradenitis suppurativa and HIV. The entire article was read only if the abstract met our inclusion criteria: English language, research papers, and studies on human population only. Papers identified as irrelevant to the topic in question as well as performed on animals or cells were excluded.

## 3. Results

As of 14 April 2022, a PubMed search of “Hidradenitis suppurativa” AND “HIV” OR “AIDS” yielded 105 articles of which 45 were removed being duplicate records, 36 were not considered because they were not relevant to the outcome of interest, not written in English, not performed on human populations only and three were excluded because they were not research papers (Figure 1). The other 21 concerning the association between HS and HIV were included in our systematic review. Nine case reports out of these twenty-one papers were studied and reviewed in their epidemiological, clinical and treatment/outcomes features (Table 1).

## 4. Discussion

### 4.1. Epidemiology

Patients with HS have multiple potential risk factors that can increase the infection rate, including epidermal disruption, immune dysregulation, and topical or oral immunosuppressive treatments [32,33]. The study conducted by Lee et al., examined the associations of HS with serious acute infections (e.g., bacterial, fungal, or viral), chronic infections (e.g., HIV, hepatitis B) and antibiotic-resistant infections [32,34,35]. The authors showed a higher prevalence of cutaneous, extracutaneous, and systemic infections in adults and children with HS than in patients with psoriasis and atopic dermatitis, which lead to increased mortality and costs of care [32,36]. Regarding the relationship between HIV and the increased rates and severity of chronic inflammatory skin disorders, such as HS, to date it remains poorly understood. Being an immunodeficiency syndrome, HIV could promote the onset and progression of HS owing to immune dysregulation [37]. Bui et al. evaluated in their study a population of patients affected by HS and HIV, assessing the rates of HS misdiagnosis in the HIV + community [37]. Data from 63 adults revealed that the onset of HS in patients with HS and HIV was later than in those living with HS only (2nd–3rd decade of life). HIV diagnosis was made before HS onset in 63.8% of subjects [37]. A predisposing effect of HIV infection in the development of HS has been suggested considering the late age of HS onset on one side and the high rate of HS onset after initial HIV diagnosis on the other [37]. Indeed, Deng et al. found a six-fold higher rate of HS diagnosis in patients with HIV compared with those without HIV [38]. However, further studies are needed to establish the exact prevalence and incidence, being both conditions largely misdiagnosed or not regarded by patients, especially in the elderly population.

### 4.2. Pathogenesis

The “primum movens” in the pathogenesis of HS is follicular hyperkeratosis that leads to rupture of the hair follicle, and subsequent inflammation of apocrine glands [1]. The presence of retained hair tracts in pseudocysts and fistulas, as highlighted at ultrasound, confirms that hair follicle is the main actor in the pathogenesis and suggests laser epilation as first approach in mild cases [39,40]. Furthermore, the presence of superinfections, although not responsible of the HS, may aggravate its clinical course [41]. However, several cytokines seem to be particularly relevant in the HS immunopathogenesis [1]. Among these, tumor necrosis factor-alpha (TNF-α) and interleukin (IL)-17 represent the key cytokines, in fact their increase correlates with disease severity [4,38]. Most notably, TNF-α levels in injured tissues of patients with HS are even higher than those found in patients with psoriasis. TNF-α leads to an increase of the T helper (Th) 17/regulatory T cells ratio, which results in an augmented production of IL-17 [4]. This last induces the expression of other cytokines such as IL-1β, IL-6, and TNF-α through a mechanism involving the nod-like receptor protein 3 (NLRP3) inflammasome [4]. Moreover, TNF-alpha might predispose to insulin resistance acting on adipocytes and muscle cells to induce insulin signaling defects and suppressing the secretion of adiponectin from adipocytes [4]. Additionally, the relationship between smoking and HS is thought to involve TNF-α. Nicotine induces infundibular epithelial hyperplasia causing follicular occlusion and rupture and, at the same time, stimulates macrophages to produce IL-1β and TNF-α and increases expression of matrix metalloproteases (MMPs). Nicotine excretion in sweat also induces TNF-α release by keratinocytes and Th17 cells [4]. There are still other cytokines involved in the pathogenesis of HS: among these it is worth mentioning IL-12 and IL-23 that play a role in the establishment of chronic inflammation [4]. IL-23 is also involved in the development and maintenance of Th17 cells [4]. Even though the precise mechanism that leads to HIV and HS association has not been clearly elucidated, it is generally agreed that HIV may activate the immune system in a persistent manner, due to the increased stimulation caused by the virion and other viral gene products [38]. Thus, during HIV infection, it is the overactivation of T cells themself that may result in an accelerated depletion of these cells through higher rates of apoptosis [38]. HIV is associated with an increased release of proinflammatory cytokines such as TNF-α, IL-6 and IL-18, this phenomenon is thought to facilitate the development of HS by promoting the expression of Th17 cells and IL-17 [38,42,43].

### 4.3. Clinical Manifestations

It is interesting to underline how atypical presentations of dermatosis and unusual associations between dermatitis characterized by extremely different pathophysiological mechanisms could suggest retroviral infection [6,24,27,28,29]. The involvement of atypical sites, such as face or thighs, must be a “wake-up call”, because it could be due to HIV-related immunosuppression, as noticed by Dhadke et al. in a middle-aged male with poor health status [28] and by Rankin et al. in a man who presented HS exclusively on his face [6]. Even the association between psoriasis and HS should alert the dermatologist to rule out or confirm a possible HIV infection, as described by Bouaddi et al. in a 45-year-old man [29]. Marfatia et al. reported a case of a 35-year-old HIV-positive man with recurrent nodular skin lesions with foul smelling discharge over face, gluteal region, thighs, and axilla. Once again, this case emphasizes how HIV-induced immunological changes alter the clinical course of chronic skin conditions as HS leading to atypical manifestations and therapeutic challenges [24], as also noticed by Manglani et al. [27]. The author highlighted a rare presentation of HS in a 13-year-old pre-pubertal HIV-infected male child who did not respond to common modalities of treatment (oral antibiotics, oral steroids, and local antibiotic wash) and had to be treated with highly active antiretroviral therapy (HAART) (zidovudine, lamivudine, and nevirapine) with no recurrences observed during follow-up visits [27]. This suggests that initiating HAART for HS should be considered as a therapeutic option in HIV-positive patients who do not respond to traditional treatments of HS [27]. HIV infection is able to considerably impair the clinical course and presentation of chronic skin diseases like HS and certain their complications, such as secondary bacterial infections [7]. Given that the majority of data related to the safety of immunosuppressants in seropositive patients comes from clinical cases, physicians should be updated about possible interactions and adverse events associated with the use of biological drugs in HS (e.g., adalimumab) in patients on HAART [44]. In addition, dermatologists should keep in mind cutaneous tuberculosis or other cutaneous mycobacterial infections on their differential diagnosis, since they may mimic the presentation of HS in HIV patients [7]. A routine full body skin examination and a multidisciplinary approach with other healthcare providers, (e.g., patient’s primary care physician and infectious disease provider), is extremely important to improve the quality patient care.

### 4.4. Histopathology

In addition to studying the cytokine landscape of HS, understanding the histopathologic alterations in this complicated disease is also of some influence. The central role of follicular occlusion in the development and progression of HS corresponds to the typical histological feature, with hyperkeratosis and hyperplasia of the follicular epithelium [45,46,47,48]. Interestingly, Boar et al. characterized the inflammatory infiltrate of HS through immunohistochemical stains (IHC), reporting a high percentage of T-cells subset in the lymphocytic cell population [49]. Like psoriasis, also HS is therefore an inflammatory skin disease associated with the activation of T-cell in which the treatments that decrease the T-cell count improve it [1,36]. Specifically, the lymphocytic infiltrate is represented by cells positive for CD3, CD4, CD8, CD68 and CD79 with a CD4+/CD8+ ratio of 2.1:1 [48,49]. From the other side, HIV infection is marked by a gradual depletion in CD4+ T-cell number and an augmented CD8+ T-cell number with an inverted CD4+/CD8+ cells ratio [36,50,51].

It could seem paradoxical that severe skin manifestations are more frequent in HIV-patients than the general population [37]. However, it is common opinion that the immune dysregulation caused by both pathologies could represent the right interpretation [36,37,51].

An interesting data was the discovery of atypical CD30-positive lymphoid cells described in common cutaneous non-neoplastic disorders including HS [52]. This finding was observed by Cepeda et al. in both HIV negative and positive patients, with a particularly prominent increase in CD30 positive cells in these latter [52]. In the setting of HIV infection, this could be referred to their upregulation [52,53]. Potential differential diagnostic problems with CD30-positive lymphoproliferative disorders could arise given that the presence of these atypical CD30+ lymphoid cells does not allow to formulate alone a diagnosis of malignancy, being they a part of the common reactive infiltrates. [52,54]. Therefore, close collaboration between oncological, dermatological, and infectious disease specialists is sometimes mandatory for a correct management of HS [29,31].

### 4.5. Therapy

The complexity of the clinical and therapeutic management of HS is burdened by a high rate of clinical failures and recurrences; more so in people living with HIV [24,26]. Depending on the severity of the disease, its extension and patients’ comorbidities, HS treatment includes topical or systemic antibiotics (such as doxycycline, clindamycin, or erythromycin), corticosteroids, isotretinoin, antiandrogens and immunosuppressors [6,24,25,26,27,28,29,30,31,55,56,57]. Severe cases of HS may also require surgical removal of the lesions, thus causing difficult to accept cosmetic alterations [25]. The combination of oral clindamycin and rifampin for 10-12 weeks has been reported as potentially beneficial in the treatment of stage 2 of HS [58]. Nevertheless, the administration of rifampin is challenging in patients receiving antiretroviral therapy. Rifampin is a potent cytochrome P450 inducer (CYP450 1A2, 2B6, 2C8/9, 2C19 and 3A4, weak inducer of CYP450 2D6). It also interacts with different transporters, inducing P-glycoprotein (PGP) and UDP-glucuronosyltransferase enzymes (UGT) and inhibiting Organic Anion Transporter (OAT) and Organic Anion Transporting Polypeptide (OATP) [59]. Unfortunately, many antiretroviral drugs share at least one of the abovementioned metabolisms and their co-administration with rifampin can compromise HIV treatment [60].

Retinoids, such as acitretin, are one of the oldest drugs in the management of HS. Their use on HIV patients should be careful as it may easily cause an increase in triglycerides and transaminases [26]. Retinoid-caused hypertriglyceridemia may also be responsible for an increase of cardiovascular risk, thus leading to different cardiovascular manifestations including myocardial infarction [26].

HS in HIV is usually more aggressive and often refractory to isotretinoin. However, Rankin et al. described the case of a 31-year-old male HIV patient with HS affecting his right jawline, who responded well to isotretinoin and intralesional triamcinolone [6].

Recently, biologic therapies have been increasingly used as treatment solutions for patients with moderate-to-severe disease. TNF-alpha inhibitors, including IgG monoclonal antibodies (e.g., adalimumab and infliximab), are the most prescribed category [57]. As TNF-alpha is a proinflammatory cytokine, its blockage ultimately leads to both reduction of inflammation and increase of susceptibility to infection [42]. However, other targets have been studied and other molecules have been used as possible treatment for HS such as: anti-IL-12/IL-23 (ustekinumab), IL-1 alpha inhibitors (anakinra and bermekimab), IL-17 inhibitors (secukinumab) and anti-IL-23 (guselkumab). They have shown to be effective in HS management even if they are not yet approved by the United States Food and Drug Administration (FDA) nor by the European Medicines Agency (EMA) [57].

As mentioned above, HIV infection can alter the clinical course of HS as seropositive patients often present with more severe cutaneous manifestations and are unresponsive to standard therapies [7].

Although biologics have already been reported as available therapeutic options for moderate to severe psoriasis [61,62] as they do not interfere with ART response, their use in HIV patients with HS is still controversial, as there is not enough scientific evidence due to the absence of randomized placebo-controlled trials. Moreover, the risk of infectious complications related to biologic therapy in already immunosuppressed patients has limited its use considerably [30,56].

Along this line, J.D. Claytor et al. decided to discontinue infliximab and methotrexate combination therapy in a 55-year-old woman with HS when an HIV test performed during the treatment came back positive. However, the immunosuppressive therapy has not been started again after she reached a good viral control and HS progressed aggressively [31].

Early in vitro studies demonstrated the involvement of TNF-alpha in HIV replication and clinical manifestations. TNF-alpha stimulates HIV expression through nuclear factor kappa beta (NFkB) pathway activation and is also a selective activator of lymphocyte T-cells. Furthermore, TNF-alpha production is high in HIV patients undergoing antiretroviral treatment and tends to increase as the infection progresses [55].

Several studies have shown a possible role of TNF-alpha in Acquired Immune Deficiency Syndrome (AIDS) manifestations such as fever, fatigue, cachexia, aphthous ulcers and dementia [56]. Thus, TNF-alpha blockade may play a role in HIV treatment. On this note, anti-TNFalpha therapy has already been administered to treat immune reconstitution inflammatory syndrome in HIV patients who failed steroid treatment [56].

Different case reports demonstrated the safety and efficacy of anti-TNF-alpha monoclonal antibodies in the treatment of HS/HIV patients, as shown in Table 1.

Even though the impact of anti-TNF-alpha on viral load and CD4+ cell count is one of the main concerns about monoclonal antibodies administration in HIV patients, only one case report describes a possible contribution of infliximab to a reduced CD4+ count and increased viral load [25].

Alecsandru at al. reported a case of a 47-year-old HIV male patient who never needed antiretroviral therapy with persistent cutaneous lesions [25]. His CD4+ T-lymphocytes were 546 cells/μL, CD8+ lymphocytes were 795 cells/μL and CD4+/CD8+ 0.6. HIV-1 viral load was 24,870 copies/mL. Tuberculin test came out negative. However, antituberculosis prophylaxis was started two months before infliximab administration. The patients soon reported an improvement of the symptomatology. After 10 months of treatment, the CD4+ cell count decreased and viral load increased, antiretroviral therapy was then initiated. No adverse event has been reported and the clinical response persisted during the maintenance therapy [25].

On the contrary, Husein-ElAhmed et al. and A. Molina-Leyva et al. described the cases of two HIV patients with refractory HS treated with anti-TNF-alpha [26,30]. Both patients were already on antiretroviral treatment and the viral load was undetectable. Husein-ElAhmed et al. administered infliximab with little improvement of skin lesions. After two weeks a recurrence was observed, and neodymium-yttrium-aluminium-garnet laser therapy was performed with partial response. Molina-Leyva et al. prescribed adalimumab treatment with a complete and maintained response at follow-ups. No impact on CD4+ or viral load was detected at follow-ups in both cases [26,30].

## 5. Conclusions

HS in HIV positive patients often presents a more aggressive course and is refractory to standard therapy compared to HIV negative patients [56]. Biological therapies have begun to be prescribed to reach disease remission in these patients. However, the efficacy and the risk related to its or other molecules administration in people living with HIV needs to be studied further.

Even if the incidence of adverse effect is low in the reviewed cases, new randomized placebo-controlled trials are still needed to be able to compare safety and efficacy of biologic therapies between seronegative and seropositive patients.

As only adalimumab is approved by FDA and EMA for HS treatment, new studies are also necessary to optimize treatment and to find new targets, thus broadening the therapeutic choices.

In conclusion, given the chance of opportunistic infection and other serious adverse effects, HIV screening should be routinely performed before biological therapy administration.

Immunosuppressive therapy could be used cautiously in patients on antiretroviral therapy with stable CD4 count and undetectable viral load at baseline, and patients should be closely monitored.

Evidence and recommendations for physicians about the management of HS in HIV positive patients are summarized in Table 2.

## Figures and Tables

**Figure 1 biomedicines-10-02761-f001:**
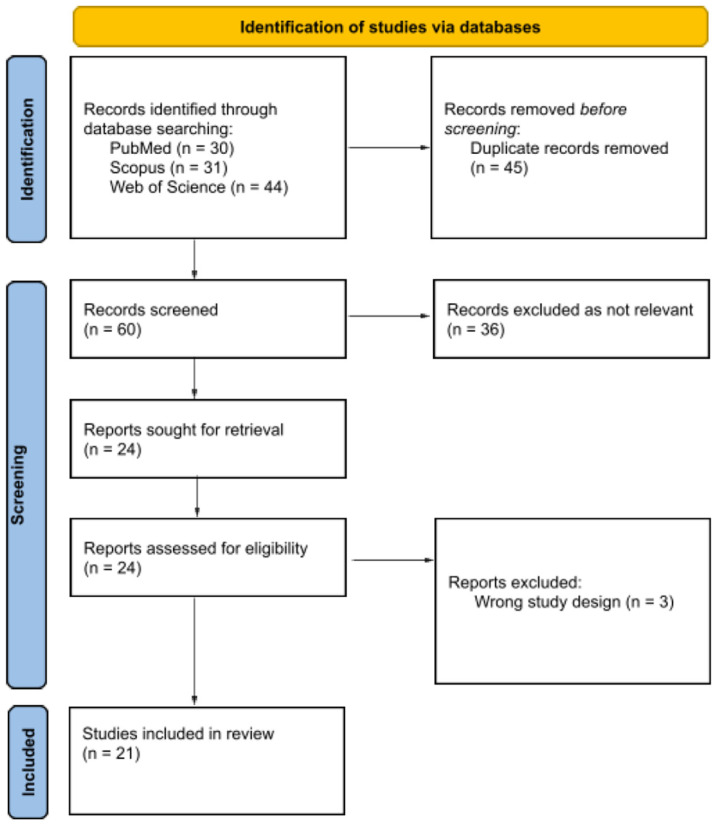
Flow diagram of studies assessed in systematic review. Diagram shows several steps of studies selection in systematic review, also describing exclusion criteria and final number of included articles for each step.

**Table 1 biomedicines-10-02761-t001:** Reported cases of hidradenitis suppurativa in individuals with HIV.

Authors, Reference Number and Year	Type of Study	Sex/Age (Year)	Duration	Location	Severity	Treatment	Outcome
**Marfatia Y et al.** [24], 2010	Case report	M/35	2 years	Face, thighs, gluteal region, and axilla	Not reported	Doxycicline, ibuprofen, dapsone	Partial response with recurrence
**Alecsandru D et al.** [25], 2010	Case report	M/47	>1 year	Axillary, gluteal, perianal, thoracic and abdominal	Severe	Infliximab, clindamycin	Remission
**Husein-ElAhmed H et al.** [26], 2011	Case report	M/47	10 weeks	Inguino-scrotal perineum and gluteal areas	Not reported	Infliximab	Partial response with recurrence
**Mangiani M et al.** [27], 2012	Case report	M/13	1 months	Right axilla	Not reported	Antiretroviral therapy (zidovudine, lamivudine, nevirapine)	Remission
**Dhadke S et al.** [28], 2016	Case report	M/“Middle-age”	10 days	Right cheek, neck, chest, shoulder, and axilla	Not reported	Local antibiotic washes, amoxicillin and clavulanic acid, prednisolone	Exitus
**Bouaddi M et al.** [29], 2017	Case report	M/45	6 months	Armpits	Mild	Not reported	Not reported
**Molina-Leyva A et al.** [30], 2018	Case report	W/39	16 weeks	Left and right axilla	Severe	Adalimumab	Remission
**Claytor J et al.** [31], 2021	Case report	F/55	“several months”	Axillary and groin	Not reported	Infliximab and methotrexate (discontinued after HIV diagnosis)	Uncontrolled after infliximab and methotrexate discontinuation
**Rankin B et al.** [6], 2021	Case report	M/31	4 months	Right jawline	Moderate	Doxycycline, Isotretinoin, intralesional triamcinolone acetonide	Remission

**Table 2 biomedicines-10-02761-t002:** Pearls for physicians: evidence and recommendations.

Pearls for Physicians	
Evidence	Recommendations
Patients with HIV may develop recalcitrant forms of inflammatory comorbidities	A multidisciplinary approach is mandatory in case of suspect cutaneous manifestations
Patients with HS have multiple potential risk factors that can increase their infection rate	As HS and HIV are largely misdiagnosed or not regarded by patients, particular attention should be paid to anamnesis and screening
HIV could promote the onset and progression of HS owing to immune dysregulation	Physicians may aware of late forms of HS
Atypical clinical presentations (and the association between physiopathologically different dermatoses) could be subtended by retroviral infection	The involvement of atypical sites should be considered as a ’wake-up call’
Clinical and therapeutic management of HS patients is characterized by a high rate of failures and recurrences, more so with HIV	Management and treatment should be as tailored as possible
First-line treatment options may be ineffective in HIV&HS patients or contraindicated because of concomitant medications	Different treatments, including surgery, must be considered, especially in most severe cases
Clinical experience seems to demonstrate that TNF-alpha inhibitors and newer biologics interfering with IL-17 and IL-23 can provide improvement in patients with HIV	Biotechnological treatments should be considered for treatment in motivated HIV&HS patients, who are monitored for CD4+ count

## Data Availability

Not applicable.

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
