# Peer review of "Hidradenitis Suppurativa in Patients with HIV: A Scoping Review"

_biomedicines, 2022, doi:10.3390/biomedicines10112761_

Round 1

Reviewer 1 Report

Authors performed a review on treatment of HS in patients with HIV. 

Manuscript is logically presented and will contribute to decision making. 

Minor comment: please specify in figure 1 that 9 of the 21 articles were selected for review. 

Author Response

Dear reviewer, 

thank you for your suggestion.

With regard the minor comment ("specify in figure 1 that 9 of the 21 articles were selected for review"), we want to point out that we have already specified it in Results with the following sentence: "The other 21 concerning the association between HS and HIV were included in our systematic review. 9 case reports out of these 21 papers were studied and reviewed in their epidemiological, clinical and treatment/outcomes features".

Thank you in advance

Best regards

Reviewer 2 Report

There are some minor comments.

1. It would be better to add clinical and histologic photos for cases of hidradenitis suppurativa.

2. It would be better to add what role follicular occlusion and secondary infection play in pathogenesis.

3. Please check English grammar and spelling.

  according with our review ->according to our review

  clavulinic acid ->clavulanic acid

  Doxycicline ->Doxycycline

4. Please check the references.

Mangiani et al. [10] in text. -> Manglani et al. [10] 

                     ? Prabhu G, Laddha P, Manglani M, Phiske M

Husein-ElAhmed et al. ->Husein-ElAhmed et al. [9]

Molina-Leyva et al. -> Molina-Leyva et al. [13]

Author Response

Dear reviewer,

thank you for your suggestion.

  1. With regard to the first point, we will add a clinical and histologic photos of hidradenitis suppurativa as soon as.
  2. We added few sentences to emphasize the role of the follicular occlusion and superinfections in the pathogenesis of HS.
  3. We checked English grammar, spelling and references according to your suggestion.

Thank you for your support

Best regards

Reviewer 3 Report

Dear Authors, thank you for performing this comprehensive review about the relationship between hiv and hs; the presence of both diseases may result in a worsening of hs, with atypical features, as suggested in your paper, and a risk for dangerous superinfections due to immunosuppression.

The introduction may be improved by adding a recent review about pathogenesis of HS (Zouboulis CC, Exp Dermatol 2020). The presence of retained hair tracts in pseudocysts and fistulas, as highlighted at ultrasound, confirms that hair follicle is the main actor in the pathogenesis and suggests laser epilation as first approach in mild cases (Wortsman X, Wortsman J. Ultrasound Detection of Retained Hair Tracts in Hidradenitis Suppurativa. Dermatol Surg. 2015 Jul;41(7):867-9 - Nazzaro G, Zerboni R, Passoni E, Barbareschi M, Marzano AV, Muratori S, Veraldi S. High-frequency ultrasound in hidradenitis suppurativa as rationale for permanent hair laser removal. Skin Res Technol. 2019 Jul;25(4):587-588) .

Furthermore, the presence of superinfections, although not responsible of the hs, may aggravate its clinical course (Benzecry V, et al. Hidradenitis suppurativa/acne inversa: a prospective bacteriological study and review of the literature. G Ital Dermatol Venereol. 2020 Aug;155(4):459-463.

page 4. this sentence needs further clarification or referencing "It is interesting to underline how atypical presentations of dermatosis and unusual associations between dermatitis characterized by extremely different pathophysiological mechanisms could suggest retroviral infection.

page 4 "Even the rare association between psoriasis and HS should alert the dermatologist to rule out or confirm a possible HIV infection, as described by Bouaddi et al. in a 45-year-old man": I do not think that the association among psoriasis and hs is "rare"; they are prevalent diseases in the population (Levin NA, Rashighi M. Psoriasis and hidradenitis suppurativa are associated with inflammatory bowel disease: a growing body of evidence. Br J Dermatol. 2022 Aug 17. doi: 10.1111/bjd.21808.  - Licata G, et al. A case of moderate hidradenitis suppurativa and psoriasis successfully treated with risankizumab. Int J Dermatol. 2022 Apr;61(4):e126-e129. - Pinter A, et al. Coprevalence of Hidradenitis Suppurativa and Psoriasis: Detailed Demographic, Disease Severity and Comorbidity Pattern. Dermatology. 2021;237(5):759-768. - Nazzaro G, Muratori S. Hidradenitis suppurativa associated with systemic lupus erythematosus and psoriasis: a therapeutical challenge. Ital J Dermatol Venerol. 2021 Dec;156(Suppl. 1 to No. 6):13-14.

page 6. "Retinoids, such as isotretinoin, are one of the oldest drugs in the management of HS": please change with acitretin, that is more indicated in the guide lines rather isotretinoin.

conclusion. this sentence "HS in HIV positive patients often presents a more aggressive course and is refractory to standard therapy compared to HIV negative patients" need a reference and an explanation (why is more aggressive and not responsive to standard drugs? the reason is that hs is more aggressive or that some drugs are contraindicated?)

"In conclusion, given the chance of opportunistic infection and other serious adverse effects, HIV screening should be routinely performed before immunosuppressive therapy administration": HIV screening should be performed before adalimumab treatment or in every case of HS or when?

Author Response

Dear Reviewer,

thank you for your suggestions.

  1. The introduction has now improved adding the reference of the review you suggested. The sentences about laser epilation and superinfections have been insert in the section “Pathogenesis”.
  2. The sentence “It is interesting to underline how atypical presentations of dermatosis and unusual associations between dermatitis characterized by extremely different pathophysiological mechanisms could suggest retroviral infection” is now clarified as references have been added.
  3. According with your suggestion, we replaced “rare” in the sentence (page 4).
  4. We changed “isotretinoin” with “acitretin”, according to the guidelines (page 6).
  5. The sentence “HS in HIV positive patients often presents…” has been improved with a reference. Cutaneous inflammatory disease, such as HS, often present with more aggressive and refractory clinical course in the presence of accompanying HIV infection and are refractory to traditional treatments when compared to seronegative patients.
  6. HIV screening should be performed before biological therapy administration.

Thank you for your support

Best regards

Reviewer 4 Report

In the manuscript entitled: Hidradenitis suppurativa in patients with HIV: a systematic review, the authors have performed a literature survey concerning the association between Hidradenitis suppurativa and HIV. This is a piece of unique information with well-documented details. 

I have a few suggestions to improve the quality of the manuscript. Please find my comments below:

The survey was performed until April 24, 2022. Authors should include if there are any new studies published after the search was performed in the revised version.

Adding a table of available therapy or clinical/pre-clinical investigational therapy for HS will be helpful. 

Page #6 Section 4.5 "Several studies have shown......." needs reference/s

Page #7: the unit microL should be designated with appropriate symbols

All the sections after the conclusion lack relevant details and need to be modified by the authors.

Author Response

Dear reviewer,

thank you for your suggestion.

  • No other new studies of interest for our review have been published after April 24, 2022.
  • According with your suggestion, we added a table “Pearls for Physicians” in which we illustrate Evidence and Raccomandation about the clinical and therapeutic management of HS.
  • In the sentence "Several studies have shown..." a reference has now been added.
  • Page #7: the unit micromL has been changed with the appropriate symbol.
  • All the sections after the conclusion have been modified with relevant details.

Thank you for your support

Best regards

Round 2

Reviewer 3 Report

no further comments